

# A Process–Based Rating Curve to model suspended sediment concentration in Alpine environments

Anna Costa[1], Daniela Anghileri[1], Peter Molnar[1]

[1]Institute of Environmental Engineering, ETH Zurich, 8093 Zurich, Switzerland

*Correspondence to*: Anna Costa (costa@ifu.baug.ethz)

**Abstract.** A Process–Based Rating Curve (PBRC) approach to simulate mean daily suspended sediment concentration (SSC) as a function of different sediment sources and their activation by erosive rainfall (ER), snowmelt (SM), and icemelt (IM) in an Alpine catchment is presented. Similarly to the traditional rating curve, the PBRC relates SSC to the three main hydroclimatic variables through power functions. We obtained the hydroclimatic variables from daily gridded datasets of
precipitation and temperature, implementing a degree–day model to simulate spatially distributed snow accumulation and snow–ice melt. We calibrated the PBRC parameters by an Iterative Input Selection algorithm to capture the characteristic response time lags, and by a gradient–based nonlinear optimization method to minimize the errors between SSC observations and simulations. We apply our approach in the upper Rhône Basin, a large Alpine catchment in Switzerland. Results show that all three hydroclimatic processes ER, SM, and IM are significant predictors of mean daily SSC (explaining
75%, 12% and 3% of the total observed variance). Despite not using discharge in prediction, the PBRC performs better than the traditional rating curve, especially during validation at the daily scale and in reproducing SSC seasonality. The characteristic time lags of the three variables in contributing to SSC reflect the typical flow concentration times of the corresponding hydrological processes in the basin. Erosive rainfall determines the daily variability of SSC, icemelt generates the highest SSC per unit of runoff, and snowmelt–driven fluxes represent the largest contribution to total suspended sediment
yield. Finally, we show that the PBRC is able to simulate changes in SSC in the past 40 years in the Rhône Basin connected to air temperature rise, even though these changes are more gradual than those detected in observations. We argue that a sediment source perspective on suspended sediment transport such as the PBRC may be more suitable than traditional discharge–based rating curves to explore climate–driven changes in fine sediment dynamics in Alpine catchments. The PBRC approach can be applied to any Alpine catchment with a pluvio–glacio–nival hydrological regime and adequate
hydroclimatic datasets.



## 1 Introduction

Sediment rating curves (RCs), based on the relation between suspended sediment concentration and discharge, have been widely applied to estimate suspended sediment load along rivers (e.g. Walling, 1974; Asselman, 1999; Lenzi and Marchi, 2000; Yang et al., 2007; Wang et al., 2008). A sediment rating curve commonly expresses suspended sediment concentration (SSC) as a power function of discharge (Q) (e.g. Campbell and Bauder, 1940; Walling, 1977). Calibration of the RC parameters is usually performed by linear regression after logarithmic transformation of the data. A correction factor is often applied (e.g. Crawford, 1991; Asselman, 2000; Horowitz, 2003; De Girolamo, 2015), to account for the underestimation of SSC induced by such transformation (Duan, 1983; Ferguson, 1986; Newman 1993). Although widely applied, RCs are usually characterized by a large variability in observations around the regression curve which can span one or more orders of magnitude, (e.g. Crawford, 1991; Walling, 1977; Asselman, 2000; Horowitz, 2003). This uncertainty is due to several reasons, most notably the fact that sediment supply is not explicitly accounted for in such relations and that the sediment flux exhibits a hysteretic behaviour with regard to instantaneous flow (Walling, 2005; Sadeghi et al., 2008; Mao and Carrillo, 2016). The premise behind this paper is that better predictions of suspended sediment concentration and yield would be achieved if factors such as the type and location of sediment sources and the sediment production and transport processes are accounted for in RC analysis (Vansickle and Beschta 1983, Syvitski, 2000; Lenzi et al., 2003; Walling, 2005; De Vente et al., 2006; Warrick, 2015). Moreover, an approach accounting for sediment sources would allow exploring potential causes of changes in suspended sediment dynamics that traditional approaches, based on discharge only, are not capable to capture. In the literature, there are several examples of alterations in suspended sediment regimes driven by changes in land use, climate or by other disturbances such as wildfire, earthquakes and flow impoundment (e.g. Loizeau and Dominik, 2000; Foster et al., 2003; Dadson et al., 2004; Yang et al., 2007; Horowitz, 2010; Costa et al., 2017). These alterations are normally addressed by calibrating different rating curve models for different sediment regimes, and by analyzing the temporal changes of the parameters (Syvitski, 2000; Yang, 2007; Hu, 2011; Huang and Montgomery, 2013; Warrick, 2015).

Sediment supply is clearly influenced by the type and location of sediment sources. Lithology and land cover influence sediment availability through soil erodibility (e.g. Stutenbecker et al., 2016), and specific sediment sources in the fluvial system (e.g. hillslopes, channels, moraines) contribute differently to sediment yield depending on their distance to the outlet and on their degree of sediment connectivity (e.g. Cavalli et al., 2013; Heckmann and Schwanghart, 2013; Bracken et al., 2015). Furthermore, it is important to know what are the triggering factors for sediment production (e.g. rainfall, overland flow, concentrated runoff) and how the produced sediment is transported downstream. For instance, rainfall erosion on hillslopes is strongly related to rainfall intensity (e.g. Van Dijk et al., 2002) and through the detachment of soil by raindrop impact and its subsequent entrainment and transport by overland flow (Wischmeier, 1959; 1978). Rainfall is also a key triggering factor for mass wasting events, such as landslides and debris flow (e.g. Caine, 1980; Dhakal and Sidle, 2004; Guzzetti, 2008), which can mobilize large amounts of fine sediment (e.g. Korup et al., 2004; Bennet et al., 2012) resulting in very high suspended sediment concentrations in the receiving streams. Snowmelt–driven overland flow may also be



important for suspended sediment production and transport from saturated soils in spring, especially in Alpine environments where abundant snowmelt can produce high hillslope runoff and be a major contributor to channel discharge (e.g. Konz, 2012). A variety of glacial erosion processes produce fine sediment in glaciated catchments (Boulton, 1974). Melt water from glaciers delivers high sediment loads which were previously stored in subglacial networks and paraglacial

environments to downstream channels (Aas and Bogen, 1988; Gurnell et al., 1996; Lawler et al., 1992) and may substantially increase suspended sediment concentration in glacially–fed streams.

Several approaches have been proposed to reduce the uncertainty associated with sediment rating curves. Some approaches attempt to indirectly account for the temporal and spatial variability of sediment supply by estimating sediment rating curves on monthly (e.g. Mao and Carrillo, 2015) or seasonal (e.g. Walling, 1974; 1977) basis, rather than annual. Other approaches

consist in calibrating rating curves specifically for different flow conditions, like rising limbs of hydrographs, etc. (e.g. Jansson, 1996; Finger et al., 2006, Anselmetti et al., 2007, De Girolamo et al., 2015), or for different precipitation intensity ranges (Guzman, 2013). A more direct way of including sediment supply is to add a supply–function to the rating curve for representing the depletion of sediment throughout the hydrological year, as proposed by Van Sickle and Beschta (1983). However, this method requires knowledge on the amount of sediment stored and the timing of sediment release, which is

usually unknown or difficult to estimate (Asselman 1999).

In this paper, we propose a different approach, which we call the process–based rating curve (PBRC), and which takes into account different sediment supply conditions by differentiating among the main erosion and transport processes typical of Alpine catchments. We consider that the suspended sediment regime is determined by sediment fluxes driven by three main hydroclimatic forcings: (1) erosive rainfall (ER), defined as liquid precipitation over snow free surfaces, which is responsible

for soil detachment and erosion along hillslopes, triggering of mass wasting events (e.g., debris flows and landslides), and enhancing channel erosion through increased discharge, (2) snowmelt (SM), which has a direct impact on hillslope erosion through overland flow, and affects channel erosion by contributing to streamflow, and (3) icemelt (IM), which transports high concentrations of fine sediment derived from the glacier bed and paraglacial areas. Due to the diversity of the erosion and transport processes (e.g. erosion driven by overland flow, soil detachment by raindrop impacts) and the variety of

sediment sources involved (e.g. hillslopes, channels, glaciers), sediment fluxes generated by these three variables (hydroclimatic forcings) are expected to contribute to suspended sediment dynamics in a complementary way, both in terms of magnitude and timing. The expectation is that partitioning suspended sediment yield into these three distinct sediment fluxes will improve SSC predictions and provide a causal explanation of SSC concentrations based on data.

The main objectives of the paper are to: (1) test the hypothesis that the three hydroclimatic variables SM, IM and ER play a

significant role in determining suspended sediment concentration in an Alpine catchment; (2) develop a model for predicting suspended sediment concentration, which accounts for the different erosional and transport processes related to SM, IM and ER without using measured discharge; (3) compare this new process–based rating curve (PBRC) with a traditional rating curve (RC) and validate its performance. The upper Rhône Basin in southern Switzerland is used as the study catchment. The upper Rhône River contributes more than 65% of the total input of particulate matter into Lake Geneva, the largest lake





in the Alps (Loizeau et al., 1997), substantially influencing the morphology and ecology of the river delta and the lake (Loizeau and Dominik, 2000; Loizeau et al., 1997). Moreover, alterations of suspended sediment concentration entering Lake Geneva have been observed in the recent past because of human impacts (Loizeau and Dominik, 2000; Loizeau et al., 1997) and changes in climatic conditions (Costa et al., 2017). Although tested only on this specific catchment, the approach

we propose here is general and can be applied in any Alpine catchment where sufficient data are available.

The paper is organized as follows: Sect. 2 describes the data pre–processing, the general hydrological modelling procedure to obtain the required hydroclimatic variables and the PBRC approach; Sect. 3 presents the upper Rhône basin and the data used in our analysis; Sect. 4 reports the main results and their interpretation; Sect. 5 concludes the manuscript summarizing and discussing the main findings.

## 2 Methods

The Rating Curve (RC) relates suspended sediment concentration to discharge through the following power function:

$$SSC_t = a_{RC} \cdot Q_t^{b_{RC}} \tag{1}$$

where $SSC_t$ is the mean daily suspended sediment concentration and $Q_t$ is the mean daily discharge.

The Process–Based Rating Curve (PBRC) we propose here relates $SSC_t$ to three hydroclimatic variables which are assumed

to mainly drive the suspended sediment regime of Alpine catchments:

$$SSC_t = a_1 \cdot ER_{t-l_1}^{b_1} + a_2 \cdot SM_{t-l_2}^{b_2} + a_3 \cdot IM_{t-l_3}^{b_3}, \tag{2}$$

where $ER_{t-l_1}$, $SM_{t-l_2}$, $IM_{t-l_3}$ are total daily basin–averaged erosive rainfall, snowmelt, and icemelt computed at time lags $l_1$, $l_2$, and $l_3$ respectively. The time lag represents the time necessary for sediment produced at a given location in the catchment to reach the outlet. In principle, the travel time depends on the sediment source location (i.e., distance from the outlet) and

the velocity of the transport (which is a function of runoff, topography, and flow resistance). Here, we assume a characteristic travel time specific of each hydroclimatic component, i.e., $l_i$ (with i = 1,2,3), which represents an average in space (i.e., over the catchment) and time (i.e., over the hydrological year). We also assume that coefficients $a_i$ and $b_i$ (with i = 1, 2, 3) may vary between the hydroclimatic variables, because they express sediment availability as well as the nonlinearity of the relation of $SSC_t$ production by each hydroclimatic variable. The PBRC does not use discharge in the

estimation of $SSC_t$.

Our methodology consists of two main steps: (1) derivation of $SSC_t$ and $ER_t$, $SM_t$, $IM_t$ datasets: mean daily $SSC_t$ from measurements of turbidity, and the hydroclimatic input variables $ER_t$, $SM_t$, $IM_t$ from spatially distributed snow and icemelt models (Sect. 2.1); (2) calibration and validation of the PBRC: we use an Input Variable Selection algorithm to calibrate $l_i$ (with i = 1,2,3) and a gradient–based optimization algorithm to calibrate $a_i$ and $b_i$ (with i = 1,2,3) (Sect. 2.2).

We then assess the model by comparing the performance of the PBRC with the traditional RC in predicting daily time series of observed $SSC_t$ in the Rhône Basin and in reproducing its long term changes (Sect. 2.3).



## 2.1 SSC and Hydroclimatic Data Modelling

This first step of the analysis is to obtain a dataset of $SSC_t$ and $ER_t$, $SM_t$, $IM_t$ for the PBRC calibration. The specific operations described here depend strongly on the data availability for the case study under consideration. In the following we describe the data analysis process for the upper Rhône Basin but we also comment about the applicability of these and

alternative operations to other catchments.

SSC sampling has been historically conducted manually, usually with low frequency (e.g., a few samples a week) and fixed intervals, because measurements are costly and time consuming (e.g. Gippel, 1995; Pavanelli and Pagliarani, 2002). This has resulted in long but intermittent SSC datasets, which are not suitable for data–driven modelling, because they might not be representative of the entire range of possible suspended sediment concentrations. On the other hand, automatic gauging

stations with optical turbidity sensors produce turbidity datasets which are continuous but usually shorter, because of the recent installation of such sensors. Because turbidity is strongly related to suspended sediment concentration (e.g. Gippel, 1995; Lewis, 1996; Pavanelli and Pagliarani, 2002; Holliday et al., 2003; Lacour et al., 2009; Métadier and Bertrand–Krajewski, 2012), the two datasets, when available at the same location, can be combined to obtain a higher frequency SSC dataset as detailed below.

Punctual manual measurements of SSC collected twice per week and continuous measures of Nephelometric Turbidity Units (NTU) are available at the outlet of the Rhône Basin. We selected measurements of NTU and SSC taken simultaneously (i.e., with a maximum time lag of five minutes). To reduce the influence of outliers, we excluded SSC and NTU data higher than the 90[th] percentile values, which correspond to 2000 mg l[-1] and 1000 NTU respectively in our dataset. Least squares regression is then used after a logarithmic transformation to fit the model:

$$SSC = a_0 \cdot NTU^{b_0} \qquad\qquad\qquad (3)$$

For the back–transformation from the logarithmic to the arithmetic scale, we applied the correction factor proposed by Duan (1983). The uncertainty of the linear regression is given by estimating 95% confidence intervals, computed as $SE \cdot \pm t_{0.025,M-2}$, where SE the standard error of mean predicted values and $t_{0.025,n-2}$ indicates the 0.975 quantile of the t–distribution with m–2 degrees of freedom, m being the sample size. Finally, we computed mean daily NTU values from

continuous measurements of turbidity, and we used the SSC–NTU relation (Eq. 3) to estimate mean daily SSC.

Observed datasets of the hydroclimatic variables $ER_t$, $SM_t$, $IM_t$ are rarely available, and would be commonly derived by hydrological modelling. The choice of the model should be driven by the data availability for calibration and the required accuracy of the simulated outputs. In our case, we use a conceptual and spatially distributed model of snow and icemelt driven by spatially distributed precipitation and temperature (Costa et al. 2017). We use gridded datasets of total daily

precipitation and mean, maximum and minimum daily air temperature to divide precipitation into rainfall and snowfall on the basis of a temperature threshold. We model ice and snow accumulation and melting with a degree–day approach (e.g., Hock, 2003). Icemelt occurs only on glacier cells that are snow–free. Likewise, rainfall is accounted as erosive only on snow–free cells. We set temperature thresholds for snow/rain division and for snow and icemelt based on the literature and





on previous studies (e.g., Fatichi et al., 2015), while we calibrate melt factors with satellite–derived snow cover (MODIS) and with discharge measured at different locations in the catchment. The threshold for distinguishing between liquid and solid precipitation is set equal to 1°C and the threshold for snow and icemelt is set equal to 0 °C. We first calibrated the snowmelt rate from snow cover maps by spatial statistics that measure the grid–to–grid matching of the model. Second, we

calibrated the icemelt rate on the basis of discharge measured at the outlet of two highly glaciated tributary–catchments. For more details on the hydrological model description and calibration see Costa et al. (2017).

## 2.2 Calibration of the PBRC

We apply the Iterative Input Selection (IIS) algorithm (Galelli and Castelletti, 2013) (1) to select which hydroclimatic variables play a significant role in predicting $SSC_t$, (2) to quantify their relative importance, and (3) to calibrate the time lags

of the sediment flux associated with each selected variable. The IIS algorithm selects the most relevant input variables, among a set of candidate input variables (in our case, mean daily $ER_{t-1}$, $SM_{t-1}$, $IM_{t-1}$ at different time lags $l$), to predict a specific output variable (in our case, mean daily $SSC_t$). It calibrates and validates a series of regression models considering different sets of input variables and selecting the ones that display the best model performances. The algorithm adopts Extremely Randomized Trees, or Extra–Trees, (Geurts et al., 2006) as regression models, because they allow dealing with

non–linear relations between input and output variables in a computationally efficient way. The Extra–Trees regression is based on a recursive splitting procedure, which partitions the dataset into sub–samples containing a specified number of elements. This splitting procedure is performed several times by randomizing both the input variable and the cut–point used to split the sample, in order to minimize the bias of the final regression (for more details see Geurts et al., 2006).

The IIS algorithm is based on an iterative procedure, which allows for the ranking of the candidate input variables according

to their significance in explaining the output variable on the basis of the coefficient of determination $R^2$ of the underlying regression model. At the first iteration, regression models are identified and the candidate variable leading to the best model performance is selected. At subsequent iterations, the original output variable (i.e. $SSC_t$) is substituted with the residual of the model computed at the previous iteration to minimize the redundancy related to the selection of input variables that are highly correlated between each other (for more details see Galelli and Castelletti, 2013). Because of the relatively short

duration of our dataset and the marked seasonal pattern that characterizes the considered candidate input variables and output variable, we randomly shuffle the dataset 100 times before running the IIS algorithm (as suggested in Galelli and Castelletti, 2013). We select parameters related to Extra–Trees based on Geurts et al. (2006) and Galelli and Castelletti (2013).

We calibrate the remaining parameters of the non–linear multivariate PBRC, i.e. $a_i$ and $b_i$ in Eq. 2, by minimizing the mean squared error (MSE) between observed and simulated SSC with a gradient–based optimization approach. We assume that

each sediment flux originates under supply–unlimited conditions, i.e. there is a positive relation between sediment transport capacity and the load of sediment mobilized and transported. Accordingly, the optimization is subject to the following constraints: $b_i > -1$ (with $i = 1,2,3$); coefficients $a_i$, (with $i = 1,2,3$) are instead not constrained, which allows for dilution





(when $a_i < 0$); and simulated $SSC_t > 0$. We repeat the optimization procedure 100 times, starting from randomly generated initial values to reduce the risk of detecting sub–optimal parameter configurations.

## 2.3 Comparison of PBRC and RC

We calibrate the parameters of the RC (Eq. 1), $a_{RC}$ and $b_{RC}$, by least squares regression applied to the logarithm of mean daily suspended sediment concentration $SSC_t$ and mean daily discharge $Q_t$. As for the SSC–NTU relation, we apply the smearing estimator of Duan (1973) to the back–transformed values of $SSC_t$ to correct for the bias (e.g., De Girolamo et al., 2015).

We then compare the ability of the PBRC to the traditional RC in reproducing mean daily $SSC_t$ time series observed at the outlet of upper Rhône Basin. The performances of the models are evaluated by computing goodness of fit measures such as coefficient of determination $R^2$, Nash–Sutcliffe efficiency NSE, and root mean squared error RMSE, over the calibration and validation periods. We compare the simulated and observed seasonal pattern of $SSC_t$ by analysing mean monthly values. In addition, we compute the mean absolute error for different percentiles of $SSC_t$ to estimate the capability of the model to reproduce high suspended sediment concentration conditions which are very important for total sediment load.

In addition, we use the PBRC and RC to simulate the time series of mean daily $SSC_t$ at the outlet of the upper Rhône basin for the 40–year period 1975–2015. We compare PBRC and RC simulation results to the 2–per–week observations of SSC on the basis of mean annual values, computed by considering only simulations corresponding to measurement days. Simultaneously to an abrupt raise in air temperature, the upper Rhône basin has experienced a statistically significant jump in mean annual SSC in mid–1980s, which has been attributed to an increase of icemelt and rainfall over snow free surfaces (Costa et al., 2017). We apply statistical tests for equality of the means on time series of mean annual SSC, simulated with the PBRC and the traditional RC, to test the ability of the models to reproduce the shift detected in the observations.

## 3 Upper Rhône Basin: description and data availability

We apply the PBRC approach on the upper Rhône Basin in the Swiss Alps (Fig. 1). The total drainage area of the catchment is equal to 5338 km$^2$ and about 10% of the surface is covered by glaciers. The basin covers a wide elevation range (from 372 to 4 634 m a.s.l.). The Rhône River originates at the Rhône Glacier and flows for roughly 170 km before entering Lake Geneva. The hydrological regime of the catchment is dominated by snow and ice melt with peak flows in summer and low flows in winter. Mean discharge is equal to about 320 m$^3$/s in summer and 120 m$^3$/s in winter, while the mean annual discharge is around 180 m$^3$/s. Basin–wide mean annual precipitation is about 1 400 mm yr$^{-1}$ and mean annual temperature is about 1.4 °C estimated at basin mean elevation.





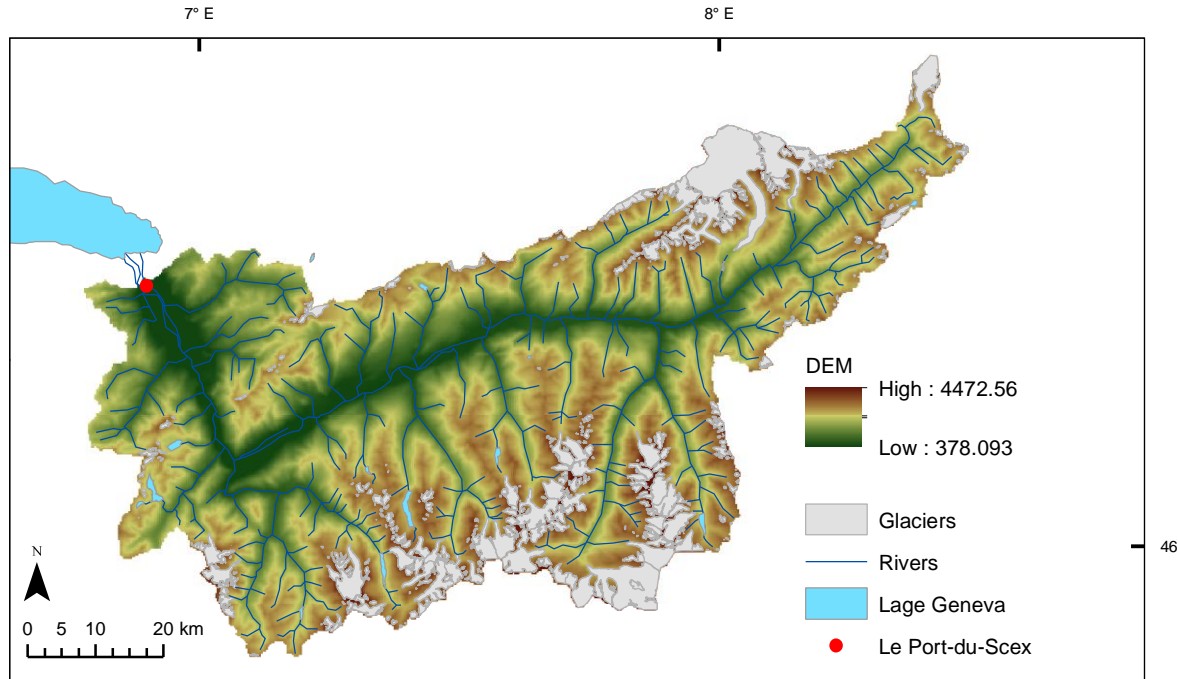

**Figure 1: Map of the upper Rhône Basin with topography, glacierized areas and river network. The measurement station le Port–du–Scex, located just upstream the Rhône River enters the Lake Geneva, is indicated with a red circle.**

5  SSC at the outlet of the catchment is characterized by the seasonal pattern typical of Alpine catchments (Fig. 2a). During winter (December – March) sediment sources are limited because a large fraction of the catchment is covered by snow and precipitation occurs in solid form. Streamflow is mainly determined by baseflow and hydropower releases (Loizeau and Dominik, 2000; Fatichi et al., 2015), and SSC assumes its minimum values. In spring, SSC increases when snowmelt–driven–runoff mobilizes sediments along hillslope and channels. Simultaneously, snow cover decreases and rainfall events

10  over gradually increasing snow free surfaces detach and transport sediment downstream, resulting in SSC peaks. In July, SSC reaches its highest values in conjunction with streamflow (Fig. 2a). In late summer (August and September), when icemelt dominates, sediment rich fluxes coming from proglacial areas maintain high values of SSC although discharge is decreasing (Fig. 2a). In terms of suspended sediment yield, low SSC conditions do not play a relevant role compared to moderate and high SSC conditions: more than 66% of the total suspended sediment load entering Lake Geneva during the

15  period May 2013 – December 2015 is estimated to be due to SSC values greater than the 90[th] percentile (Fig. 2b).

Porte–du–Scex is a measurement station at the outlet of the Rhône River into Lake Geneva (Fig. 1), where the Swiss Federal Office of the Environment (FOEN) collects discharge, SSC and turbidity data. Mean daily discharge is available since 1905, while SSC is measured twice per week since October 1964. Quality–checked continuous measurements of NTU are available since May 2013 (Grasso et al., 2012).



(a)

(b)

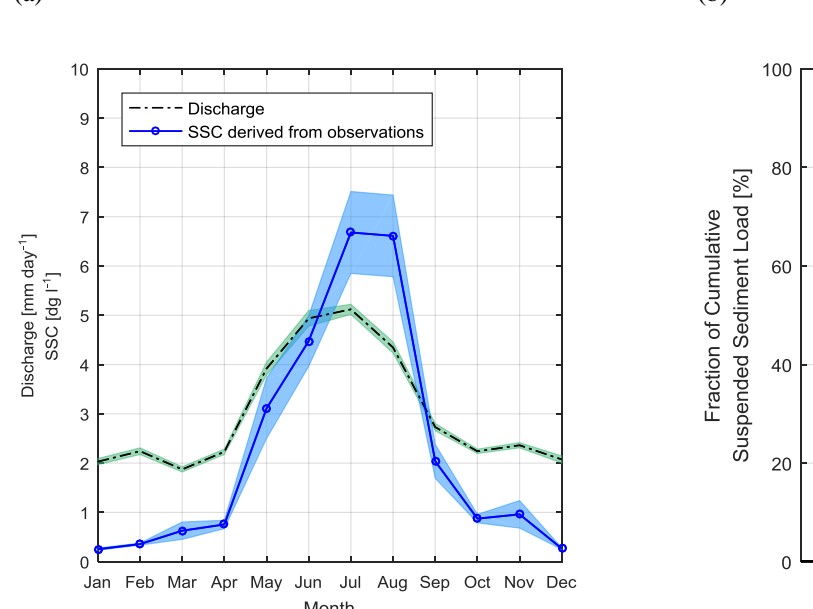

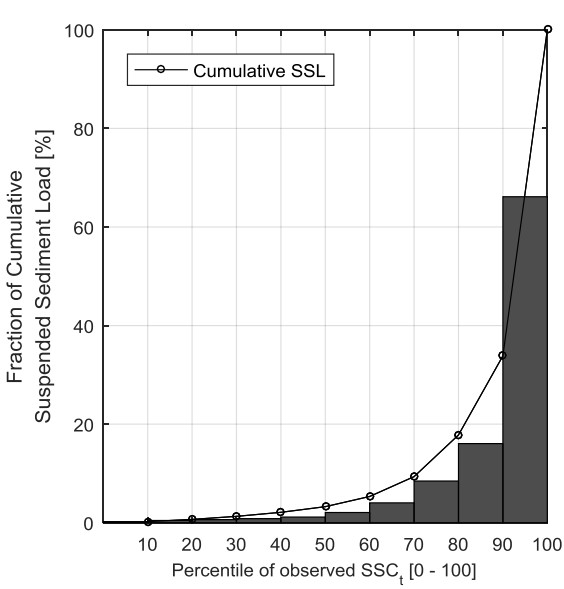

**Figure 2: (a) Mean monthly values of: discharge measured at the outlet of the catchment (dash–dot black line), $SSC_t$ derived from observations of NTU (solid blue line with circles). Coloured shaded areas represent the range corresponding to ± standard error. Mean values and standard errors are computed over the entire observation period: from 01 May 2013 to 31 December 2015. (b) Cumulative suspended sediment load (SSL) transported at the outlet of the upper Rhône basin during the observation period as function of different percentiles of $SSC_t$ (black line with circles). Bars represent the fraction of the total SSL transported by the different percentiles of $SSC_t$ (e.g.: more than 66% of total SSL is transported with $SSC_t > 90^{th}$ percentile).**

We estimate the hydroclimatic variables for the forty year period 1975–2015 with the spatially distributed degree–day model of snow and icemelt. For this we use a DEM with a spatial resolution of 250×250 m (Federal Office of Topography – Swisstopo). For the climatic dataset, we use gridded total daily precipitation, mean, maximum and minimum daily air temperature at ~ 2×2 km resolution provided by the Swiss Federal Office of Meteorology and Climatology (MeteoSwiss). These datasets are produced by spatial interpolation of quality–checked measurements collected at meteorological stations (Frei et al., 2006; Frei, 2014). Snowcover maps used for the calibration of the snowmelt rate were derived for the period 2000–2008 in a previous study (Fatichi et al., 2015) from the 8–day snow cover product MOD10A2 retrieved from the Moderate Resolution Imaging Spectroradiometer (MODIS) (Dedieu et al., 2010). We consider the GLIMS Glacier Database of 1991 to define the initial configuration of the ice covered cells. To calibrate the icemelt rate, we use mean daily discharge data measured at the outlet of two highly glacierized tributary catchments: the Massa and the Lonza (Costa et al., 2017). When applying the IIS algorithm (Sect. 2.2), we consider mean daily $ER_{t-l}$, $SM_{t-l}$, $IM_{t-l}$ at time lags $l$ from 1 day to 7 days. This choice was driven by the size of the basin and the expected flow concentration times in the basin. We calibrate the PBRC on data from the period 1 May 2013 – 30 April 2015 (730 days) and validate it over the period 1 May 2015 – 31 December 2015 (245 days). For the sake of comparison, calibration and validation periods are the same also when considering RC.





## 4 Results and Discussion

### 4.1 Calibration of the SSC–NTU relation

The linear model fitted to the logarithm of SSC and NTU satisfactorily represents the variance of the process, with a coefficient of determination $R^2$ equal to 0.93 (Fig. 3). The calibrated parameters are $a_0 = 0.591$ and $b_0 = 1.237$, after applying the correction factor for back–transforming from logarithmic to arithmetic scale. The 95% confidence interval representing the uncertainty of the regression for mean predicted values is reasonably narrow close to average NTU values (SSC = $109.10 \pm 6.74$ mg/l at mean NTU), and, as expected, increases towards higher turbidity conditions (SSC = $816.09 \pm 88.98$ mg/l at 90th percentile of NTU).

We are aware that the relation between SSC and turbidity: (1) is site–specific, (2) may vary seasonally as function of discharge and transported grain sizes, and (3) depends on sediment sources, because the size, the shape, and the composition of suspended material may influence values of turbidity (Gippel, 1995). For this reason, in this analysis: (1) we apply a site–specific SSC–NTU relation, (2) we calibrate the relation over a wide range of NTUs and discharge conditions to account for the seasonal variability in grain sizes transported by the flow, and (3) we derive the SSC–NTU relation based on a relatively short period of time, in which there is not evidence of changes in sediment sources. In addition, by allowing a non–linear relation between SSC and NTU, we take partially into account the variability of turbidity with grain size. Higher suspended sediment concentrations are expected to transport proportionally larger grains, and the exponent in the SSC–NTU relation was expected to be greater than 1.

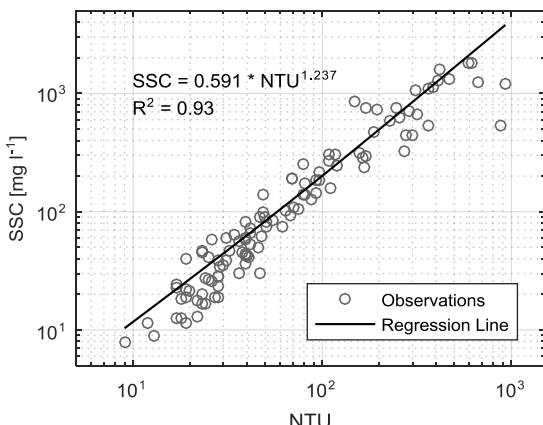

**Figure 3: Scatterplot of NTU and SSC observed simultaneously (i.e. with a maximum lag of 5 minutes) at the outlet of the catchment (grey circles), the calibrated regression of Eq. 3 (black line).**





## 4.2 Iterative Input Variable Selection Algorithm: selected variables and characteristic time lags

The IIS algorithm selects erosive rainfall at 1 day lag $ER_{t-1}$, snowmelt at 2 days lag $SM_{t-2}$, and icemelt at 5 days lag $IM_{t-5}$ as the most relevant variables to predict mean daily $SSC_t$ (Fig. 4a). We consider only the first 3 selected variables because the cumulative explained variance, expressed as coefficient of determination $R^2$, is greater than 0.9 and the contribution of additional variables is negligible (Fig. 4b). The IIS result is interesting for several reasons.

First, it confirms our hypothesis that erosion and transport processes driven by all 3 hydroclimatic variables ER, SM and IM play a role in determining the suspended sediment dynamics of the Rhône Basin, and likely in most Alpine basins with pluvio–glacio–nival hydrological regimes.

Second, it gives an indication on the relative importance of the different processes. In fact, the contribution of each hydroclimatic variable to the overall $R^2$ differs quite significantly. While $ER_{t-1}$ explains more than 75% of the variability of $SSC_t$, the melting components $SM_{t-2}$ and $IM_{t-5}$ are responsible for a much lower fraction of the variance, i.e. 12% and 3% respectively (Fig. 4b). Figure 3a shows also that the selection of the $ER_{t-1}$ as most relevant explanatory variable is unambiguous, because the variable is always selected as first in the 100 repetitions of the algorithm (Fig. 4a). Instead, it is clear that the relative importance of snowmelt–driven processes (always ranked as second most relevant variable) is greater than the one of icemelt (always ranked as third most relevant variable), although the selection of the time lags is not always the same in the 100 repetitions of the algorithm. These results are in accordance with the physical processes underlying the erosion and sediment transport dynamic. The higher intensity that characterizes rainfall events in comparison to the melting components is more likely to generate peaks of SSC. In accordance, ER is responsible for a large fraction of the process variability. Indeed, intense rainfall events can detach and mobilize large amounts of sediment by the impact of raindrops on the soil and by precipitation–driven runoff along hillslopes (Wischmeier, 1959; 1978). The sharp rise in streamflow, which typically follows a precipitation event, results in an increase in transport capacity that may further entrain sediment previously stored along channels. Precipitation is also one of the main triggering factors of mass wasting events, like landslides and debris flow (e.g. Caine, 1980; Dhakal and Sidle, 2004; Guzzetti, 2008), in which large quantities of sediment may be instantly released to the river network (e.g. Korup et al., 2004; Bennet et al., 2012). Conversely, the slow and continuous effect of snowmelt–driven runoff on hillslope and channel erosion contributes to the seasonal pattern of SSC and plays a secondary role in explaining its daily variability. Similarly, the process governing icemelt–driven erosion and sediment transport is continuous and more gradual than the one characterizing erosive rainfall. In addition, while the input of SM may be relevant for a relatively long period of time (March – September), the contribution of IM is substantial only during summer months (July – September), which can explain its lower explanatory power relative to SM.

The third interesting result is related to the representative time lags selected for the different sediment processes. The algorithm identifies $ER_{t-1}$ in 100% of the runs; $SM_{t-2}$ in 60% of the runs and $SM_{t-1}$ in 37% of the runs; and, $IM_{t-5}$ in 62% of the runs and $IM_{t-6}$ in 14% of the runs. The selected time lags, which represent basin–averaged mean travel times of sediments from their source to the outlet of the catchment, including also the time required to produce runoff sufficient to




entrain sediment, are again in agreement with the physical processes of erosion and transport in the catchment. The response of the catchment to rainfall is relatively fast: runoff along hillslopes is generated almost instantaneously, streamflow in channels rises, and the travel time of sediment is relatively short (1 day). Snowmelt is instead generated mostly at higher elevations. The associated sediment fluxes are therefore characterized by a longer travel time on the average (1–2 days).

5   Finally, icemelt fluxes originate at the very upstream headwater catchments where glaciers are located. Accordingly, travel times of sediment fluxes show the longest time lag (5–6 days). Due to seasonality and the elevation–temperature dependence, the distance between the outlet of the basin and the origin of SM and IM fluxes increases gradually during the melting season. This results in a non–unique selection of time lags for SM and IM. Conversely, the spatial distribution of rainfall presents a much less marked seasonal pattern, which results in a unique characteristic time lag.

(a)

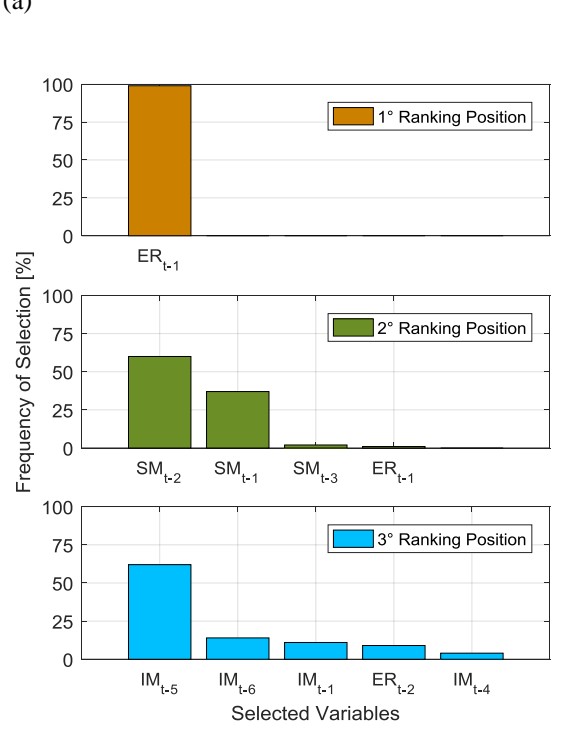

(b)

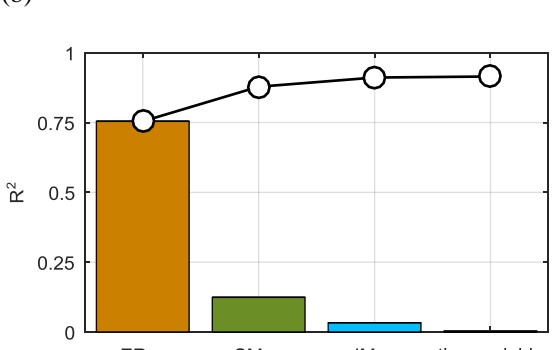

**Figure 4: Results of the IIS algorithm (100 runs over the period 01 May 2013 – 30 April 2015): (a) frequency of the variables selected as the most relevant variables in predicting $SSC_t$, ranked according to descending explanatory power $R^2$; (b) fraction of the variance of $SSC_t$ ($R^2$) explained by the most frequently selected explanatory variables ($ER_{t-1}$, $SM_{t-2}$, $IM_{t-5}$), and cumulative explained variance (black line with circles).**

### 4.3 Calibration of the PBRC

After the calibration of the remaining PBRC parameters (see Sect. 2.2), the PBRC takes the following form:

$$SSC_t = 0.429 \cdot ER_{t-1}^{1.212} + 0.120 \cdot SM_{t-2}^{1.745} + 1.256 \cdot IM_{t-5}^{1.321} \tag{4}$$

where $SSC_t$ is expressed in dg $l^{-1}$ and $ER_{t-1}$, $SM_{t-1}$, and $IM_{t-5}$ are expressed in mm day$^{-1}$.

15  The values of the parameters indicate that IM generates the greatest contribution to $SSC_t$ per unit volume of water. This is in agreement with the fact that meltwater originated in glaciated areas is characterized by very high SSC (Gurnell et al., 1996;





Lawler et al., 1992). Dilution and sediment dis–connectivity, due to the large size of the catchment and the long distance between glaciers and the outlet, may attenuate the contribution of icemelt–driven sediment fluxes to the total sediment yield of the upper Rhône Basin. Nevertheless, parameters of the PBRC reflect the strong contribution of glacier–fed fluxes to $SSC_t$ reaching the outlet of the catchment. Parameters also show that for a given amount of volume of water, fluxes driven by ER

carry higher SSC than fluxes generated by SM. This is in accordance with the higher unit erosional power of rainfall due to its higher intensity and to the effect of raindrop impact on soil detachment. However, in terms of total daily suspended sediment load ($SSL_t$), estimated here as the product of mean daily $SSC_t$ simulated with the PBRC and measured mean daily discharge $Q_t$, SM clearly provides the greatest contribution, followed by ER and finally IM. In Fig. 5 we show the relative contribution to total daily $SSL_t$ of ER, SM and IM over the 2–year calibration period, in the form of empirical cumulative

distribution functions of the relative contribution of the three terms in Eq. 4 driven respectively by $ER_{t-1}$, $SM_{t-2}$ and $IM_{t-5}$.

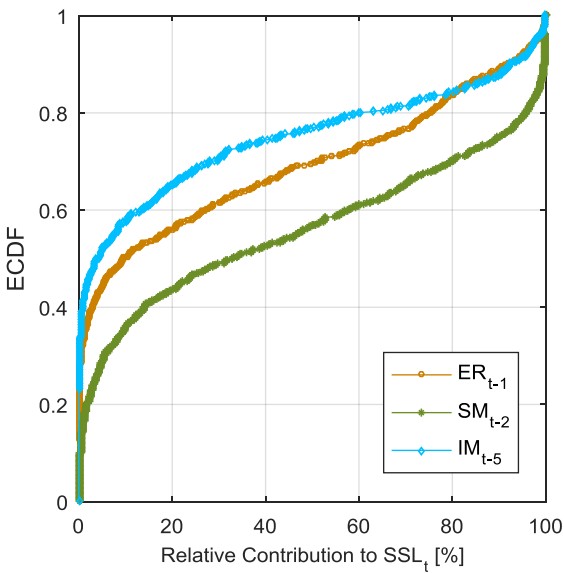

**Figure 5: Empirical cumulative distribution function over the 2–year period 01 May 2013 – 30 April 2015 of the relative contribution to total daily suspended sediment load $SSL_t$ of the three terms of the PBRC (Eq. 4) driven by $ER_{t-1}$, $SM_{t-2}$ and $IM_{t-5}$. $SSL_t$ is computed as the product of mean daily $SSC_t$ simulated with PBRC and measured mean daily discharge $Q_t$.**

**4.4 Comparison of PBRC and RC**

The calibrated RC results in the following form:

$$SSC_t = 0.080 \cdot Q_t^{2.634} \tag{5}$$

where $SSC_t$ is expressed in dg l$^{-1}$ and $Q_t$ is expressed in mm day$^{-1}$. These values are in agreement with parameters of suspended sediment rating curve used in a previous study on the upper Rhône basin (Loizeau and Dominik. 2000).

Table 1 compares the performances of the PBRC and RC in reproducing mean daily observed $SSC_t$ as measured by the coefficient of determination $R^2$, Nash–Sutcliffe efficiency NSE, and root mean squared error RMSE, over the calibration and



validation periods. The PBRC and the RC show similar performances over the calibration period, e.g. NSE is above 0.6 in both cases, despite the fact that the PBRC does not use observed discharge in the estimation of $SSC_t$. Furthermore, the performance of the RC drops in the validation period (e.g. NSE equal to 0.38), while the PBRC retains satisfactory performance (e.g. NSE equal to 0.63).

Figure 6 contrasts the PBRC and the RC with SSC derived by observations of NTU in terms of mean monthly values. It clearly shows that PBRC is better than the RC in reproducing the seasonal sediment dynamics in all seasons except in autumn. The traditional RC, instead, significantly overestimates SSC in late spring and beginning of summer (May – June), anticipates the SSC peak, and underestimates SSC in August and September. Conversely, mean monthly values of SSC predicted by PBRC are satisfactorily similar to observations especially when the amount of sediment transported in

suspension is at its highest values (July – August). The overestimation of SSC simulated with the PBRC in early spring (March, April) and especially early autumn (September, October) is most likely related to an overestimation of the SM component. This bias can be ascribed to the intrinsic simplification of melt processes in degree–day models, and to the fact that infiltration is neglected. The traditional RC overestimates SSC in winter because it relies on streamflow only, thus accounting for flows coming from hydropower reservoirs, which are most likely poorer in sediments (Loizeau and Dominik,

2000). The PBRC, instead, by taking into account the physical processes determining SSC can account for the limitation of sediment supply by natural processes typical of winter months, but it also cannot simulate sediment retention by hydropower.

**Table 1. Goodness of fit measures for the PBRC and the traditional RC in calibration (left) and validation (right): coefficient of**
**determination ($R^2$), Nash–Sutcliffe efficiency (NSE), root mean squared error (RMSE), and mean absolute relative error for values larger than $SSC_t > 90^{th}$ percentile (MARE ( $SSC_t > 90^{th}$)).**

| | Calibration 01.05.13 – 30.04.15 | | Validation 01.05.15 – 31.12.15 | |
|---|---|---|---|---|
| | PBRC | RC | PBRC | RC |
| $R^2$ | 0.65 | 0.61 | 0.63 | 0.38 |
| NSE | 0.65 | 0.61 | 0.63 | 0.38 |
| RMSE [dg l$^{-1}$] | 2.77 | 2.93 | 3.50 | 4.53 |
| MARE ($SSC_t > 90^{th}$) [dg l$^{-1}$] | 0.43 | 0.51 | 0.38 | 0.53 |






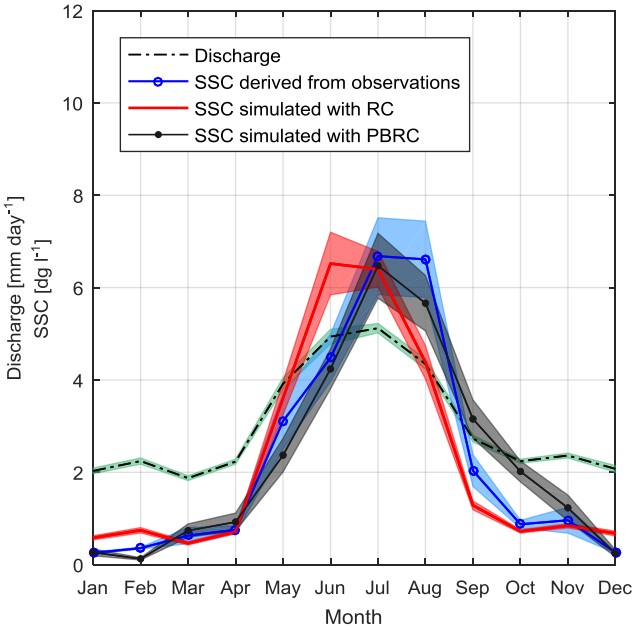

**Figure 6: Mean monthly values of: discharge measured at the outlet of the catchment (dash–dot black line), SSC$_t$ derived from observations of NTU (solid blue line with circles), SSC$_t$ simulated with the traditional RC (solid red line), and with the PBRC (solid black line with dots). Coloured shaded areas represent the range corresponding to ± standard error. Mean values and standard errors are computed over the entire observation period (01 May 2013 to 31 December 2015).**

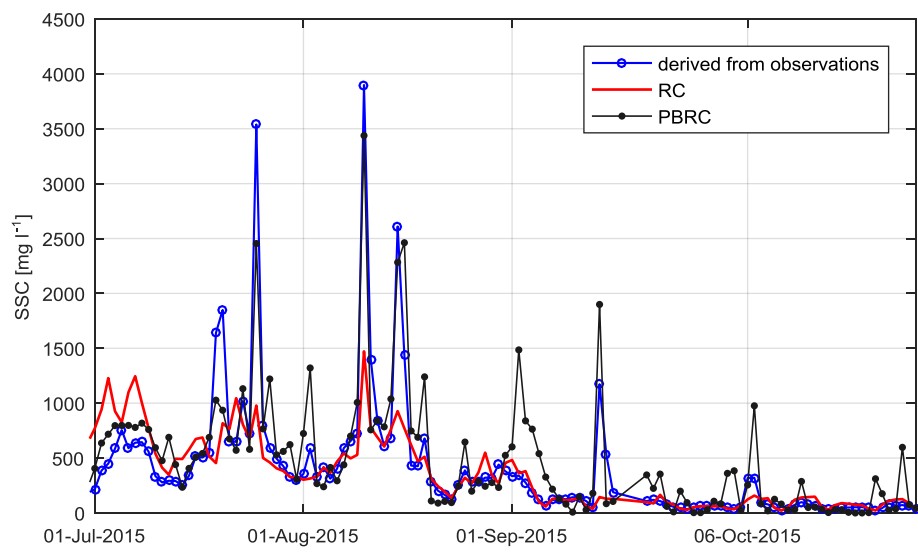

**Figure 7: Time series of mean daily SSC$_t$ for the period 01 July 2015 – 30 October 2015: derived from observations of NTU (solid blue line with circles), simulated with the traditional RC (solid red line) and with the PBRC (solid black line with dots).**





The differences between PBRC and RC are visible also at the daily scale in Fig. 7. It is evident that the PBRC outperforms the RC in reproducing the observed $SSC_t$ peaks in July and August. This is also well represented by the Q–Q plot in Fig. 8 and by the MARE–90[th] performance metric in Table 1, i.e. the mean absolute relative error for $SSC_t$ values greater than the 90[th] percentile, computed as the ratio between the mean absolute error and the mean observed value. This is a valuable

feature of the PBRC because high sediment concentrations are responsible for most of the total sediment yield.

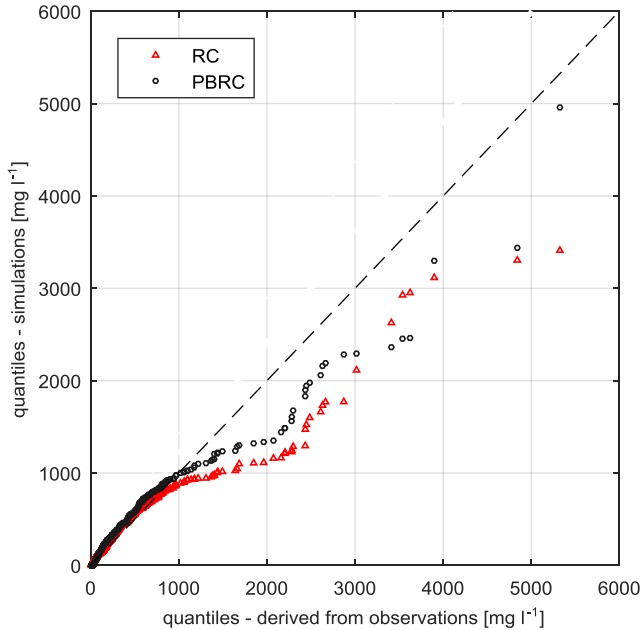

**Figure 8: Q–Q plot of mean daily $SSC_t$ over the observation period (01 May 2013 – 31 December 2015) simulated with the traditional RC (red triangles) and with the PBRC (black circles).**

We applied the traditional RC and the PBRC to simulate 40–year long time series of mean daily SSC at the outlet of the upper Rhône basin to evaluate the capability of the two models in simulating long–term dynamics of suspended sediment. We explored (1) if a change in mean SSC that occurred in the Rhône Basin in the mid–1980s due to an increase in air temperature (Costa et al., 2017) can be simulated by any/both methods and (1) if the 2–per–week SSC sampling is leading to a loss in the detectability of this change.

Figure 9a shows mean annual values of SSC for the period 1975–2015: from 2–per–week observations (top), and from simulations with the traditional RC (middle) and with the PBRC (bottom). We computed mean annual simulated SSC values by PBRC and RC only from days when observations were taken, to make a fair comparison with observed values. None of the two models can reproduce the change in the observed mean annual SSC satisfactorily. However, the PBRC shows a higher $R^2$ between observed and simulated time series ($R^2 = 0.51$ as opposed to $R^2 = 0.30$ for RC). A two–sample two–sided

t–test for equality of the mean does reveal a statistically significant jump (5% significance level) in mean annual simulated





SSC in mid–1980s by PBRC and not by RC, but this is only if the actual time of the change is known a priori (Costa et al., 2017). In fact, even for the PBRC prediction, the change seems more gradual than a shift in the mean.

(a)                                                                                          (b)

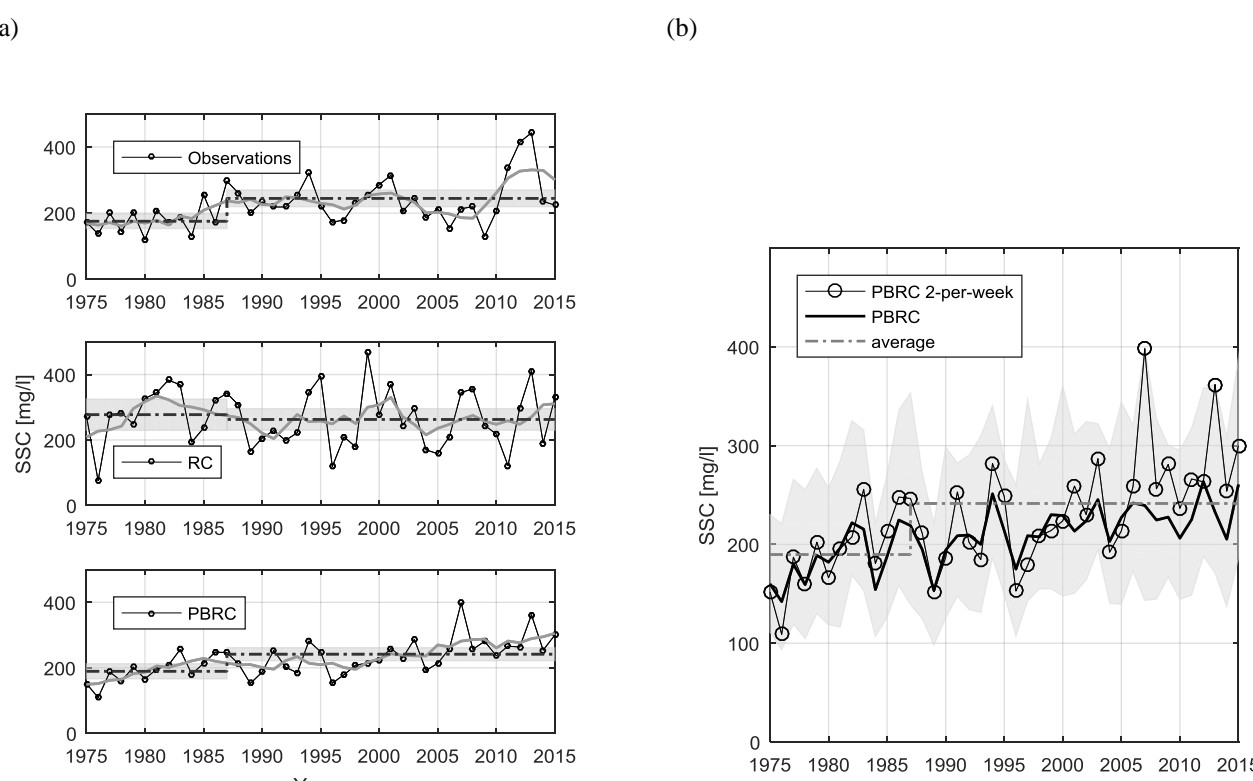

**Figure 9: (a) Time series of mean annual SSC$_t$ for the 40–year period 1975–2015, estimated from 2–per–week: observations (top), simulations with the traditional RC (middle) and with the PBRC (bottom). Dash–dot lines represent average values before and after mid–1980s. Uncertainty related to mean estimates is shown with gray shaded area (± 1.96 · standard error). (b) Time series of mean annual SSC for the 40–year period 1975–2015, computed based on: daily PBRC simulations (bold black line), daily PBRC simulations only corresponding to measurement days (black line with circles). For the prior time series, average values before and after mid–1980s are shown with a dash–dot grey line. Uncertainty related to the 2–per–week selection (1000 random selections) is represented by the shaded light grey area.**

To examine if the procedure of SSC sampling could partly contribute to the low detection of change in mean SSC we compare mean annual SSC simulated with PBRC computed on the basis of: (1) all simulation days (bold black line), (2) only simulation days corresponding to sampling days (black line with circles), (3) 1000 random selections of two simulated values per week (grey shaded area) (Fig 9b). The upper and lower boundaries of the grey shaded area, represent the yearly highest and lowest mean values obtained by choosing randomly (1000 random selections) two SSC$_t$ values per week from the time series of mean daily SSC$_t$ simulated with the PBRC. Due to the limited number of measurements per year (104 on average), and the high variability which characterizes suspended sediment dynamics, the time series of mean annual values





differ significantly among the random selections (Fig. 9b). A gradual change in SSC is detectable in all time series, but a simple shift in the mean is not. This indicates that punctual measurements, collected twice per week, may be inadequate to properly capture seasonal and intra–annual variability in SSC and a higher sampling resolution is needed for better capturing the dynamics of SSC.

**5 Conclusions**

In this paper, we develop a Process–Based Rating Curve (PBRC) approach to simulate mean daily suspended sediment concentration (SSC) in Alpine catchments by differentiating the potential contributions of erosional and transport processes typical of Alpine environments, i.e. (1) erosive rainfall (ER) defined as liquid precipitation over snow free surfaces, (2) snowmelt (SM), and (3) icemelt (IM). While the traditional rating curve (RC) expresses SSC as a power function of

streamflow, the PBRC expresses SSC as the sum of three power functions, each related to total daily, basin–averaged $ER_{t-l_1}$, $SM_{t-l_2}$ and $IM_{t-l_3}$. We obtained the hydroclimatic variables $ER_t$, $SM_t$, $IM_t$, by using a conceptual spatially distributed model of snow accumulation, snow and ice melt driven by precipitation and temperature at daily resolution. We calibrated the PBRC parameters by applying the Iterative Input Selection (IIS) algorithm (to calibrate the characteristic time lags $l_1$, $l_2$, and $l_3$.) and by minimizing the mean squared error (MSE) with a gradient–based optimization approach (to calibrate the

remaining parameters). We compared the ability of the PBRC to the traditional RC in reproducing daily SSC time series observed in the upper Rhône basin, a large Alpine catchment in Switzerland.

Our main findings are summarized as follows. (1) All the three hydroclimatic processes ER, SM, and IM are significant predictors of mean daily SSC at the outlet of the upper Rhône basin, explaining respectively 75%, 12% and 3% of the total observed variance. (2) The characteristic time lags of the three variables in contributing to SSC are: 1 day for ER, 2 days for

SM, and 5 days for IM, which are time lags connected with the typical flow concentration times of these hydrological processes distributed spatially in the basin. (3) Although ER is responsible for the greatest fraction of the variability of SSC, coefficients of the PBRC indicate that IM generates by far the greatest contribution to SSC per unit of water volume and SM contributes the most in terms of total suspended sediment load. These results are in agreement with the expectations and data from past studies of rainfall, snowmelt and icemelt in the catchment (e.g. Costa et al., 2017). (4) The PBRC is capable of

reproducing the pattern of SSC even though it does not include discharge in the model. Although the PBRC and RC perform similarly in simulating observed SSC over the calibration period, the PBRC performs substantially better than traditional RC in validation at the daily scale, and in capturing seasonality, especially in summer when SSC are highest. This is particularly relevant because more than 66% of the total suspended sediment load reaching the outlet of the upper Rhône basin in the observation period is transported by SSC values larger than the 90[th] percentile. (5) The PBRC is able to detect changes in

SSC in the past 40 years, while the RC is not. However, these changes are more of a gradual nature than a shift in the mean such as was detected in observations in the mid–1980s by Costa et al. (2017). The 2−per−week manual sampling of SSC is potentially masking a detailed recognition of change in SSC dynamics in the past.



In summary, the results suggest that a more process–based approach in predicting suspended sediment concentrations, which accounts for sediment sources and transport process driven by erosive rainfall, snowmelt and icemelt, instead of only/purely discharge, may significantly improve SSC modelling in Alpine catchments, especially when the purpose is to analyse climate–induced changes in sediment dynamics. Although these results are specific for the upper Rhône basin, the approach

is general, and may be employed in other Alpine catchments with pluvio–glacio–nival hydrological regimes where sufficient data are available.

**Author contribution**

A. Costa, D. Anghileri and P. Molnar designed the methodology. A. Costa and D. Anghileri developed the code and carried out simulations and computations. All co–authors contributed to the manuscript. The authors declare that they have no

conflict of interest.

**Acknowledgements**

We thank the Federal Office of the Environment (FOEN) for providing discharge, suspended sediment concentration and turbidity data. We also thank Alessandro Grasso (FOEN) for the explanation on the SSC and turbidity measurement procedures. This research was supported by the Swiss National Science Foundation Sinergia grant 147689 (SEDFATE).

Daniela Anghileri was supported by the Swiss Competence Centre on Energy – Supply of Energy (SCCER–SoE).

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
