# Peer review of "A Process-Based Rating Curve to model suspended sediment concentration in Alpine environments"

_Hydrology and Earth System Sciences, 2017_

## Referee Comment (RC1) · Anonymous Referee #1 · 28 Jul 2017

In the submitted paper authors propose a new method for the estimation of the daily suspended sediment concentration (SSC). The method is based on the relationship among SSC and three variables, namely the total daily basin-averaged erosive rainfall, snowmelt and icemelt. These three variables are estimated based on the daily gridded datasets and model results. The authors in the proposed method (PBRC) do not directly use daily discharge as one of the factors influencing the SSC. The best characteristics of the response time lags are estimated using the Iterative input selection methodology. The proposed model is compared to the traditional type of sediment rating curve model. The comparison is made using the data from the Alpine catchment in Switzerland. The results indicate that the proposed methodology yields better es-

timates of the SSC than traditional rating curve and it is able to better reproduce the seasonal variability in the sediment transport.

The paper is well written and the presented topic is in the scope of the HESS journal. The paper also presents a novel concept for the SSC estimation (to the best of my knowledge) and the methodology used is clearly described. The language is understandable and the text is readable. I only have next comments/suggestions:

Page 4, Equation 1: arc and brc should be defined.

Page 5, lines 17-18: A statistical test could be used to confirm this assumption (e.g., Grubb's test for outliers). Otherwise, selection of this threshold seems arbitrary. Alternatively, authors should additionally describe their decision.

Page 5, line 23: Replace "SE the" with "SE is the".

Page 6, lines 3-4: You could mention, which spatial statistics were used.

Page 9, line 19: More discussion about the catchment time of concentration could be added.

Page 10, lines 3-4: Corresponding p-value could be added to the R2.

Page 11, section 4.2: I find this discussion very interesting. My question is: would SSC estimation results using just ERt-1 variable be much worse than using all three parameters? Additional calculations are needed in order to derive the IM and SM values. Thus, what is the trade-off between model complexity (adding additional variables) and estimation results? Authors could make a comparison or expand the discussion about this.

Page 11, line 12: Probably Figure 4a and not Figure 3a?

Page 13, lines 1-3: Sediment connectivity could be estimated using the SedInConnect tool that was developed by Cavalli et al. (2013) (reference is also cited in the submitted paper) and is available at: https://github.com/HydrogeomorphologyTools or

http://www.sedalp.eu/download/tools.shtml since DEM is available. Thus, you could confirm this hypothesis.

Page 14, Table 4: Besides these criteria you could also check the descriptive statistics of residuals because these can sometimes reveal additional information.

Page 16, lines 18-19: What about other goodness-of-fit criteria?

Page 18, line 17-19: Could this observation be confirmed with some statistical test or could maybe additional analysis proposed under comment Page 11, section 4.2 be performed?

Pages 18-19, Conclusions: Some general conclusion could also be added about the complexity of tested methods.

---

## Referee Comment (RC2) · Anonymous Referee #2 · 18 Aug 2017

Comments to the Author Summary of the manuscript This manuscript (ms) presents a Process–Based Rating Curve (PBRC) to estimate suspended sediment transport in a Swiss Alpine River. PBRC estimates suspended sediment concentration by computing the sum of three rating curves (RC), using instant rain (ER), snow melt (SM) and ice melt (IM) instead of discharge. ER, SM and IM is estimated using gridded datasets and degree day factors. While temperature thresholds and ice melt factors are adopted from previous studies (Fatichi et al. 2015, Costa et al. 2017), respectively, melt factors are calibrated with MODIS maps. The PBRC equation is then calibrated using an iterative input selection algorithm. The results reveal that PBRC improves the estimation of daily SSC considerably, compared to RC based on discharge only (see Fig. 7). The

study concludes that: i) ER, SM and IM contribute to SSC, ii) the time lag for the 5338 km2 large catchment is only 1day, 2 days and 5 days for ER, SM and IM, respectively, iii) IM contributes most per unit water, iv) PBRC reproduces daily SSC better than RC, and v) long term sediment loads are better estimated by PBRC than RC.

Evaluation In summary I think that a process based approach, as developed in this study, is needed to estimate SSC in Alpine catchments. Accordingly, I do think that the topic of the study is relevant. However, the authors fail to address a major issue in the Rhone valley: hydropower operations. Runoff of most glacier-fed tributaries of the Rhone are governed by hydropower operations leading to four major alterations of SSC: i) melt and runoff water is detained in reservoirs, ii) suspended sediment get trapped in reservoirs, iii) SSC in outflows of reservoir is almost constant, as it is mostly composed of glacial silt and iv) periodic flushing of reservoirs lead to exceptionally high SSC. All of the four impacts mentioned above are due to hydropower operations, and accordingly independent of ER, SM or IM. Nevertheless, I agree with the authors that PBRC provides a better estimation of SSC than RC. Accordingly, I recommend that the authors reflect on the impacts of hydropower operations on SSC and revise the methodology and manuscript accordingly. I leave it up to the editors and the readers of HESS to decide if this revision can be done in the frame of major revisions or should rather be done in the frame of a new ms.

Prior to publications I recommend: 1) Addressing the impacts of hydropower operations on SSC in the Rhone valley. This has been investigated by numerous authors, some of which are cited but not correctly put in context in the current ms. How much of the winter discharge comes from hydropower reservoirs? How much smaller is the discharge in summer due to storage in reservoirs? How does this affect SSC? 2) The estimation of ER, SM and IM is complex and in my opinion should be addressed using a hydrological model, taking into account the complexity of the Rhone valley. I find it inconsistent to adopt some model parameters (e.g. temperature threshold and ice melt parameters) from previous studies and calibrate other parameters to direct observations. Perhaps

this can be addressed within a sensitivity analysis. 3) I recommend to compare total suspended load estimations with methods described in other studies and published by regional and federal agencies, e.g. FOEN. Furthermore, an uncertainty analysis would be very helpful. I also would like to see a plausibility check of all major conclusions: can rain be responsible for 75% of SSC, how does this compare to other studies? Is a time lag of 1, 2 and 5 days realistic, how does this compare to flood peaks after heavy precipitation events? How much is delayed in hydropower reservoirs? How does the IM contribution compare to other studies? 4) I recommend to avoid mass referencing (e.g. pg2, ln15, 6 references are listed) but to be more specific why references are relevant. Three relevant references are sufficient to fortify a statement. I recommend to select only directly relevant references and build on previous works. 5) Figure 4 and 9: why are there three panels for one heading (or letter, I would recommend adding a heading) on the left and only one heading for one panel on the right? I would present annual loads rather than SSC, this would make your study more relevant for future studies. 6) I recommend shortening the text. 7) Finally, I recommend to add a reflection why this study is needed and how it complements previous studies. I would also recommend to start the abstract and introduction by introducing the problematic of high sediment loads, rather than jumping directly to the methods.
* * *

---

## Author Comment (AC1) · 21 Aug 2017

We thank Referee #1 for her/his helpful review. We have analysed the suggestions and we report in the following our response to the major comments.

1. Comment 2: Page 5, lines $17-18$: A statistical test could be used to confirm this assumption (e.g., Grubb's test for outliers). Otherwise, selection of this threshold seems arbitrary. Alternatively, authors should additionally describe their decision.

We agree with Referee #1 that we should comment more extensively on the rationale behind the threshold selection, and we acknowledge that the term "outliers" (page 5,

line 17) is misleading. We removed SSC and NTU observations larger than the 90th percentile (corresponding to 2000 mg/l and 1000 NTU respectively) because we doubt the representativeness of such high measurements for the cross−section, due to the sampling procedure, which is punctual in space and in time, and due to possible measurement errors at high NTUs. For example, since SSC and NTU measurements are not taken exactly at the same point in space and in time, a short and highly concentrated suspended sediment pulse, due to the entrainment of fine sediment close to the measurement station, could be detected by one of the two sensors only. Nevertheless, following the suggestion of the Referee we applied the Grubbs' test to detect statistically significant outliers. We first log−transformed the SSC and NTU data to obtain a distribution as close as possible to a normal one, and second, we applied the test. At 5% significance level, the Grubbs' test does not identify outliers. We, thus, applied our methodology (computation of SSC−NTU relation, IIS algorithm and calibration/validation of the PBRC and the traditional RC) on the entire dataset, that is without excluding any high values. As shown in Fig. 1, the goodness−of−fit measures (coefficient of determination ($R^2$), Nash–Sutcliffe efficiency (NSE), root mean squared error (RMSE), mean absolute relative error for values larger than the 90th percentile (MARE (SSC(t) > 90th)), and skewness of the residuals ($\lambda$residuals)) are very similar to the ones reported in the paper, meaning that excluding the data that we do not consider representative is not changing the results significantly. We will revise the manuscript to clarify the reason why we prefer to remove high values from the dataset and explain the test we have conducted to verify the effect.
* * *
2. Comment 7: Page 11, section 4.2: I find this discussion very interesting. My question is: would SSC estimation results using just ERt−1 variable be much worse than using all three parameters? Additional calculations are needed in order to derive the IM and SM values. Thus, what is the trade−off between model complexity (adding additional variables) and estimation results? Authors could make a comparison or expand the

discussion about this.

We thank Referee #1 for this comment. We agree that it is indeed interesting to evaluate the performance of the PBRC taking as the predictor only erosive rainfall at 1 day lag, ER(t-1). However, to compute ER, defined as liquid precipitation over snow free areas, it is necessary to model snow cover and so snowmelt (SM). One option to evaluate the performance of the SSC estimation without modelling SM and IM, is to calibrate the PBRC with the single predictor liquid precipitation, R. We analysed this option as well. Results of the IIS algorithm confirm that the characteristic time lag for rainfall R in the upper Rhone basin is equal to 1 day. After calibration, the model, which we call here rainfall−RC takes the following form (Eq. 1): SSC(t) = 0.787 · R(t-1)^0.978.

Although the performance of the rainfall−RC (Eq. 1) are lower than for the original PBRC, it performs satisfactorily especially in validation (Fig. 2), because the model can capture SSC peaks (Fig. 3b and Fig. 4). This result is not surprising because, as discussed in the manuscript, rainfall is responsible for large part (75%) of the variability in SSC(t). However, the new model, based on R(t-1) only, substantially underestimates medium and low values of SSC(t) (Fig. 3b and 2). This is particularly evident when looking at mean monthly values, especially in summer (June − August), when snow and icemelt largely contributes to runoff and suspended sediment load (Fig. 3a). In addition, the simulated SSC is equal to zero every time it does not rain, which is obviously an artefact of the model structure. So overall, such a model is not very satisfactory. We agree with Referee #1 that a discussion of the trade−off between model complexity and performance could be included in the revised manuscript. In Figure 5, we propose a sketch to qualitatively compare different approaches for predicting SSC (on the matrix columns) in terms of model complexity and model suitability for representing SSC features (on the matrix rows). We consider four modelling approaches to simulate mean daily SSC(t) at the outlet of an Alpine catchment, ordered in terms of model simplicity: the traditional RC, where the streamflow Q is the only predictor, the rainfall−RC with liquid precipitation R as the only predictor, the PBRC proposed in the paper, and

spatially distributed and physically based models of erosion and transport of sediment. To rank the model performance, we focus on 4 main features: the capability of capturing the seasonal pattern of SSC and the peaks of SSC, the sensitivity to climatic conditions and to the activation/deactivation of different sediment sources. Among the four models, the traditional RC and the rainfall−RC are the simplest because they are fully data−driven. They can both reproduce the time series of SSC(t), with better performances in calibration for the traditional RC and in validation for the rainfall−RC. As discussed in the manuscript, rainfall is mainly responsible for SSC peaks, due to its intense nature. Therefore, the rainfall−RC captures better the peaks of SSC than the traditional RC. However, the traditional RC captures the seasonal pattern of SSC better than the rainfall−RC, because the latter reproduces SSC only during rainy days, while RC reflects the seasonal pattern of streamflow. The traditional RC can be sensitive to changes in climatic conditions only if such changes directly influence discharge, which is not always the case (e.g., Costa et al., 2017), and it is insensitive to possible alterations of sediment sources being based solely on streamflow at the outlet of the basin. Conversely, rainfall−RC is clearly directly linked to changes in the precipitation regime, and so it can be partially sensitive to possible alterations of sediment sources. The PBRC is more complex than the RC and the rainfall−RC, because it requires to model snow accumulation, and snow and ice melting but it significantly improves predictions of SSC, both for peak values and seasonality. Moreover, PBRC is sensitive to climate induced changes in sediment dynamics and can partially (indirectly) account for alterations of sediment sources. Spatially distributed, physically based models have potentially higher modelling power than the PBRC and the RCs, and are in principle characterized by higher sensitivity both to changes in climatic conditions and alterations of sediment sources. On the other hand, they are characterized by much higher complexity both in representing the erosional and transport processes and in routing sediment fluxes to the outlet. They also require considerable effort to be calibrated and a significant amount of data to be used. In summary, we believe that the PBRC represents a good compromise between model performance and complexity (Fig. 5).

—————————————————————————————————————————

3. Comment 9: Page 13, lines 1−3: Sediment connectivity could be estimated using the SedInConnect tool that was developed by Cavalli et al. (2013) (reference is also cited in the submitted paper) and is available at: https://github.com/HydrogeomorphologyTools or http://www.sedalp.eu/download/tools.shtml since DEM is available. Thus, you could confirm this hypothesis.

Thanks to the reviewer's comment, we realized that the sentence at page 12, lines 1−3 is misleading and we will remove it from the revised manuscript. The PBRC presented in the paper is indeed a lumped model where the spatial component is only partially accounted for by the time lags characteristic of the three hydroclimatic variables. We are currently working on a spatially distributed model based on the PBRC, in which we will account for sediment connectivity by applying the sediment connectivity index developed by Cavalli et al. (2013), as suggested by the Referee #1. Results of this on−going work will be presented in a separate manuscript.

—————————————————————————————————————————

4. Comment 10: Page 14, Table 4: Besides these criteria you could also check the descriptive statistics of residuals because these can sometimes reveal additional information.

In the revised manuscript, we will add the skewness of the residuals (Figure 6). Results indicate that residuals of neither the PBRC nor the traditional RC are normally distributed. However, for the traditional RC residuals are more negatively skewed than for the PBRC.

—————————————————————————————————————————

5. Comment 12: Page 18, line 17−19: Could this observation be confirmed with some statistical test or could maybe additional analysis proposed under comment Page 11,

section 4.2 be performed?

See reply to bullet point n. 2 (reviewer's comment n. 7).
* * *
6. Comment 13: Pages 18−19, Conclusions: Some general conclusion could also be added about the complexity of tested methods.

See reply to bullet point n. 2 (reviewer's comment n. 7).

| | PBRC (outliers removed) | **PBRC (entire dataset)** | RC (outliers removed) | **RC (entire dataset)** |
|---|---|---|---|---|
| | | Calibration | | |
| $R^2$ | 0.65 | **0.66** | 0.61 | **0.62** |
| NS | 0.65 | **0.66** | 0.61 | **0.62** |
| RMSE [dg l$^{-1}$] | 2.77 | **1.02** | 2.93 | **1.07** |
| MARE(> 90th) [dg l$^{-1}$] | 0.43 | **0.42** | 0.51 | **0.50** |
| $\lambda_{residuals}$ | −3.59 | **−3.26** | −5.00 | **−4.82** |
| | | Validation | | |
| $R^2$ | 0.63 | **0.63** | 0.38 | **0.39** |
| NS | 0.63 | **0.63** | 0.38 | **0.39** |
| RMSE [dg l$^{-1}$] | 3.50 | **1.29** | 4.53 | **1.66** |
| MARE(> 90th) [dg l$^{1-1}$] | 0.38 | **0.37** | 0.53 | **0.54** |
| $\lambda_{residuals}$ | −2.14 | **−2.15** | −3.68 | **−3.58** |

**Fig. 1.** Goodness of fit measures for the PBRC and the traditional RC in calibration and validation , on the entire dataset and after removing SSC and NTU observations larger than 2000 mg/l and 1000 NTU.

|  | Calibration | | | Validation | | |
|---|---|---|---|---|---|---|
|  | PBRC | **PBRC with only** $R_{t-1}$ | RC | PBRC | **PBRC with only** $R_{t-1}$ | RC |
| $R^2$ | 0.65 | **0.47** | 0.61 | 0.63 | **0.60** | 0.38 |
| NS | 0.65 | **0.44** | 0.61 | 0.63 | **0.54** | 0.38 |
| RMSE [dg l$^{-1}$] | 2.77 | **3.50** | 2.93 | 3.50 | **3.90** | 4.53 |
| MARE(> 90$^{th}$) [dg l$^{-1}$] | 0.43 | **0.57** | 0.51 | 0.38 | **0.55** | 0.53 |
| $\lambda_{residuals}$ | −3.59 | **−3.16** | −5.00 | −2.14 | **−1.37** | −3.68 |

**Fig. 2.** Goodness of fit measures in calibration and validation for the PBRC, the traditional RC and the rainfall−RC with R(t-1) as only predictor.

[Figure]

(a)

(b)

**Fig. 3.** (a) Mean monthly values of measured discharge and SSC derived from observations of NTU, and simulated with RC, PBRC and rainfall−RC. (b) Q−Q plot of mean daily SSC simulated with the three models.

[Figure]

**Fig. 4.** Time series of mean daily SSC derived from observations of NTU and simulated with the traditional RC, the PBRC, and the rainfall$-$RC (01 July 2015 $-$ 30 October 2015).

| | RC(Q) | rainfall-RC(R) | PBRC | Spatial and Physical |
|---|---|---|---|---|
| Simplicity | | | | |
| Sediment sources | | | | |
| Climate | | | | |
| SSC Seasonality | | | | |
| SSC Peaks | | | | |

Sensitivity to

Capability to reproduce

high

medium

low

**Fig. 5.** Sketch representing the trade−off between model complexity and performance.

|  | Calibration | | Validation | |
|---|---|---|---|---|
|  | PBRC | RC | PBRC | RC |
| $R^2$ | 0.65 | 0.61 | 0.63 | 0.38 |
| NS | 0.65 | 0.61 | 0.63 | 0.38 |
| RMSE [dg l$^{-1}$] | 2.77 | 2.93 | 3.50 | 4.53 |
| MARE(> 90$^{th}$) [dg l$^{-1}$] | 0.43 | 0.51 | 0.38 | 0.53 |
| $\lambda_{\text{-residuals}}$ | **−3.59** | **−5.00** | **−2.14** | **−3.68** |

**Fig. 6.** Goodness of fit measures in calibration (left) and validation (right) for the PBRC and the traditional RC.

---

## Author Comment (AC2) · 23 Aug 2017

We thank Referee #2 for her/his helpful review. We have analysed the suggestions and we report in the following our response to the comments and give indications on how we will revise the manuscript.

1) Addressing the impacts of hydropower operations on SSC in the Rhone valley. This has been investigated by numerous authors, some of which are cited but not correctly put in context in the current ms. How much of the winter discharge comes from hydropower reservoirs? How much smaller is the discharge in summer due to storage in reservoirs? How does this affect SSC?

[Figure]

We are aware that hydropower operations affect the flow and sediment regime of the upper Rhone basin, through flow and sediment impoundment and flow regulation, which results in a substantial decrease of discharge in summer and increase in winter. Our empirical model partially accounts for the effect of hydropower operations on SSC magnitude and timing because we calibrated the parameters of the PBRC using the observed time series of SSC at the outlet of the basin, which are impacted by the hydropower operations. Therefore, the coefficients (a1, a2, a3), the exponents (b1, b2, b3) as well as the time lags specific of each hydroclimatic variable (l1, l2, l3) include the impacts of reservoirs and hydropower operations (e.g. delays in sediment transfer are accounted for in the time lags chosen by the IIS algorithm). Even though our approach is relatively simple and cannot capture all the complexities of the sediment dynamic of the upper Rhone basin, the results show that the PBRC, which accounts for hydropower operations only indirectly through the model parameters, performs better that the traditional RC, which accounts more directly for the hydropower regulation because it accounts directly for discharge. We will make this point clearer in the revised manuscript. Because we consider the suggestion of the Referee #2 valuable, we will also test an alternative PBRC model which distinguishes among the contribution to SSC of the area of the catchment in natural conditions and the area impacted by hydropower operations.
* * *
2) The estimation of ER, SM and IM is complex and in my opinion should be addressed using a hydrological model, taking into account the complexity of the Rhone valley. I find it inconsistent to adopt some model parameters (e.g. temperature threshold and ice melt parameters) from previous studies and calibrate other parameters to direct observations. Perhaps this can be addressed within a sensitivity analysis.

The aim of the PBRC approach is to relate SSC to the potential sediment sources and fluxes represented by the three hydroclimatic forcings in the basin and not to discharge which is the outcome of the basin water balance. This is a data−based approach which

uses a simple temperature−based model to simulate ER, SM and IM and require less data to be calibrated and used than most of the available hydrological models. The results of the calibration and validation show that, although simple, the model to estimate ER, SM and IM performs satisfactorily both in space (snow cover) and time (see Costa et al., 2017). In past work we have in fact used a hydrological model to simulate the Rhone Basin streamflow including all relevant hydrological processes (Fatichi et al., 2015), but this includes additional model−related uncertainties which in this work we aim to avoid. However, thanks to the reviewer's comment, we realized that we should better clarify how we calibrated the model for ER, SM and IM. In fact, the model was consistently calibrated on observed data (more details in Costa et al., 2017) and not adopting parameters values from previous studies as suggested by the Referee. Moreover, we performed a sensitivity analysis on the parameters of the snow model, more specifically on the snowmelt factor and the temperature thresholds for snow accumulation and melt (see Costa et al., 2017, Supplementary Material). We will discuss some of the results of the sensitivity analysis and we will describe more extensively the calibration procedure in the revised manuscript, in order to avoid possible misunderstandings.
* * *
3) I recommend to compare total suspended load estimations with methods described in other studies and published by regional and federal agencies, e.g. FOEN. Furthermore, an uncertainty analysis would be very helpful. I also would like to see a plausibility check of all major conclusions: can rain be responsible for 75% of SSC, how does this compare to other studies? Is a time lag of 1, 2 and 5 days realistic, how does this compare to flood peaks after heavy precipitation events? How much is delayed in hydropower reservoirs? How does the IM contribution compare to other studies?

In the revision we will compare the mean annual suspended sediment load from SSC predicted by our model and observed/derived from observations, together with other estimates that we can find, including uncertainty in the mean. We agree with Referee

**2 that it would be valuable to also compare other modelling results with previous works. We have compared IM with previous studies and we will discuss this in the revised manuscript. As for the other suggested comparisons, we are not aware of any study which distinguishes among the relative contributions of the different sources to SSC. However, we will look further into the literature and, if any, we will include in the revised manuscript references to comparable works. Finally, as already commented in the manuscript, we believe that the time lags automatically identified by the iterative input selection algorithm are realistic, considering the process involved in suspended sediment production and transport and the extent (mean flow pathway length) of the catchment. It should be remembered, though, that these time lags represent average values in space, over the entire catchment, and in time, thus accounting for both dry and wet conditions as well as for meteorological events of different intensities which are peculiar of the different seasons. We could partially account for the effect of flood peaks by including time varying parameters in the PBRC formulation or by spatially explicit computation, which we are testing in future research. A more complex formulation which accounts for monthly varying parameters was tested, but the improvement in the results was not significant to justify the increased number of parameters (from 6 of the presented PBRC to 6*12 of the time varying formulation).**
* * *
4) I recommend to avoid mass referencing (e.g. pg2, ln15, 6 references are listed) but to be more specific why references are relevant. Three relevant references are sufficient to fortify a statement. I recommend to select only directly relevant references and build on previous works.

In the revised manuscript, we will review the references, in order to avoid mass referencing.
* * *
5) Figure 4 and 9: why are there three panels for one heading (or letter, I would recommend adding a heading) on the left and only one heading for one panel on the right? I would present annual loads rather than SSC, this would make your study more relevant for future studies.

We agree with Referee #2 that the panels heading can confuse the reader, and we will adjust the figures accordingly. Regarding suspended sediment load estimations, see reply to comment number 3.

————————————————————————————————————————

6) I recommend shortening the text.

We will try to identify the parts of the manuscript which can be shortened without compromising the information content and the clarity of the paper.

————————————————————————————————————————

7) Finally, I recommend to add a reflection why this study is needed and how it complements previous studies. I would also recommend to start the abstract and introduction by introducing the problematic of high sediment loads, rather than jumping directly to the methods.

We intend to strengthen the comparison between this and other approaches for modelling SSC and making clear the added−value of this work, as a data−based empirical analysis of potential suspended sediment sources as they are reflected in hydroclimatic triggering variables. Based also on the comments of Referee #1, we will discuss this point in the revised manuscript (see our Reply to the Referee #1, bullet point n. 2 and Fig. 5). In addition, we will mention both in the abstract and the introduction why the estimation of suspended sediment concentration and load are relevant.

———————————————————————————

---

## Referee Comment (RC3) · T. Steenhuis (Referee) · 6 Sep 2017

The authors in this manuscript obtain a well fitting sediment concentration rating curve by introducing four additional fitting parameters above the two that are required for the original simple power relationship. They use then some fancy fitting routines and find as expected a better fit to the observed sediment concentration data than the two parameter model. To make the method to work they need one of the most complete data sets that exist in the world. They conclude that in the future we should collect similar data sets to get better estimates of sediment concentrations.

Before this manuscript can be published in HESS, it will require several improvements.

[Figure]

A great number of abbreviations are introduced, making the article almost impossible to read. Abbreviations are not explained in the figures. Figures should stand alone and abbreviations should be mentioned. Headings of sections have abbreviations. Just too much jargon.

There should be some previous research done on Alpine sediment fluxes and processes. The authors do not mention any of these research studies in the following statement.

"In this paper, we propose a different approach, which we call the process–based rating curve (PBRC), and which takes into account different sediment supply conditions by differentiating among the main erosion and transport processes typical of Alpine catchments. We consider that the suspended sediment regime is determined by sediment fluxes driven by three main hydroclimatic forcings: (1) erosive rainfall (ER), defined as liquid precipitation over snow free surfaces, which is responsible for soil detachment and erosion along hillslopes, triggering of mass wasting events (e.g., debris flows and landslides), and enhancing channel erosion through increased discharge, (2) snowmelt (SM), which has a direct impact on hillslope erosion through overland flow, and affects channel erosion by contributing to streamflow, and (3) icemelt (IM), which transports high concentrations of fine sediment derived from the glacier bed and paraglacial areas. Due to the diversity of the erosion and transport processes (e.g. erosion driven by overland flow, soil detachment by raindrop impacts) and the variety of sediment sources involved (e.g. hillslopes, channels, glaciers), sediment fluxes generated by these three variables (hydroclimatic forcings) are expected to contribute to suspended sediment dynamics in a complementary way, both in terms of magnitude and timing. The expectation is that partitioning suspended sediment yield into these three distinct sediment fluxes will improve SSC predictions and provide a causal explanation of SSC concentrations based on data."

I doubt very much that the signal of raindrop impact is preserved somewhere hundreds of kilometers down. Pick up of sediment in rills of plowed soils and deposition afterwards could completely overwhelm the raindrop impact signal (Moges et al., 2016). Moreover in mountainous environment saturation excess rainfall dominates and total rainfall explains better the runoff amounts and sediment concentrations than the loads than the intensity (Tilahun et al., 2013, 2015). Unlike what the authors write in their manuscript it is the total rainfall in Guzman et al (2013) that is related to the sediment concentrations in the Ethiopia highlands and not the rainfall intensity (page 3 around line 10). I do not know how the Alpine environments are different from the Ethiopia highlands, but that it is the task of the authors to research the processes that are really occurring in the Alpine watersheds.

It is interesting that in the response to the reviewer 2 the authors write.

"Our empirical model partially accounts for the effect of hydropower operations on SSC magnitude and timing because we calibrated the parameters of the PBRC using the observed time series of SSC at the outlet of the basin, which are impacted by the hydropower operations. Therefore, the coefficients (a1, a2, a3), the exponents (b1, b2, b3) as well as the time lags specific of each hydro-climatic variable (l1, l2, l3) include the impacts of reservoirs and hydropower operations (e.g. delays in sediment transfer are accounted for in the time lags chosen by the IIS algorithm). Even though our approach is relatively simple and cannot capture all the complexities of the sediment. . .."

This contradicts the earlier explanation above about the processes in the watershed. In other words the authors present a model fitting routine that is inspired by some of kind reality but reality has ultimately very little to do with the explanation of the results. For example, what would happen to the sediment concentration when the reservoir operation changes? This cannot be simulated by the sediment rating curve model.

It is well known from the equifinality approach of (Beven and Freer, 2001) that many different combinations of parameters give a best fit for the signal at the outlet. The authors found this best fits using 6 parameters. There could be a range of parameter sets that give the same result. Moreover the model chosen might not be the best. It

should be addressed how optimum is the fitting parameter set.

This review is not to indicate that the manuscript cannot be published ultimately. However any fancy explanations about what happens in the watershed based on the fitting of the sediment concentrations seems out of place unless it can be shown based on experimental evidence that the parameters in the watershed affect the concentration at the outlet. Simply in my opinion, the manuscript is about a model with six parameters that is fitting the output of some kind of simple hydrological model to very detailed measurements of daily sediment concentration. The authors should be realistic what can be done with the model and where it can be used for. For example how is the management of the hydropower dams been changed over the years and can that be the reason that the sediment concentration are changing?

Reviewer: Tammo Steenhuis

References Beven KJ and Freer J. 2001. Equifinality, data assimilation, and uncertainty estimation in mechanistic modelling of complex environmental systems, Journal of Hydrology, 249, 11–29.

Moges MA, Zemale, FA, Alemu, ML, Ayele GK, Dagnew DC, Tilahun SA and Steenhuis TS 2016 Sediment concentration rating curves for a monsoonal climate: upper Blue Nile, SOIL 2: 337-349

Tilahun SA, Guzman CD, Zegeye AD, T. A. Engda TA, Collick AS, Rimmer A and Steenhuis TS. 2013. An efficient semi-distributed hillslope erosion model for the sub humid Ethiopian Highlands. Hydrology and Earth System Sciences 17: 1051-1063

Tilahun SA, Ayana EK, Guzman CD, Dagnew DC, Zegeye AD, Tebebu TY, Yiftaru B, Steenhuis TS. 2016. Revisiting storm runoff processes in the upper Blue Nile basin: The Debre Mawi watershed. CATENA 143:47-56